# Maternal Pre-Existing Diabetes: A Non-Inherited Risk Factor for Congenital Cardiopathies

**DOI:** 10.3390/ijms242216258

**Published:** 2023-11-13

**Authors:** Stéphanie Ibrahim, Bénédicte Gaborit, Marien Lenoir, Gwenaelle Collod-Beroud, Sonia Stefanovic

**Affiliations:** 1Aix Marseille University, INSERM, INRAE, C2VN, 13005 Marseille, France; stephanie.ibrahim@univ-amu.fr; 2Department of Endocrinology, Metabolic Diseases and Nutrition, Pôle ENDO, APHM, 13005 Marseille, France; 3Department of Congenital Heart Surgery, La Timone Children Hospital, APHM, Aix Marseille University, 13005 Marseille, France; 4Aix Marseille University, INSERM, Marseille Medical Genetics, 13005 Marseille, France

**Keywords:** pregestational diabetes, cardiogenesis, congenital heart defects, genetics and epigenetics

## Abstract

Congenital heart defects (CHDs) are the most common form of birth defects in humans. They occur in 9 out of 1000 live births and are defined as structural abnormalities of the heart. Understanding CHDs is difficult due to the heterogeneity of the disease and its multifactorial etiology. Advances in genomic sequencing have made it possible to identify the genetic factors involved in CHDs. However, genetic origins have only been found in a minority of CHD cases, suggesting the contribution of non-inherited (environmental) risk factors to the etiology of CHDs. Maternal pregestational diabetes is associated with a three- to five-fold increased risk of congenital cardiopathies, but the underlying molecular mechanisms are incompletely understood. According to current hypotheses, hyperglycemia is the main teratogenic agent in diabetic pregnancies. It is thought to induce cell damage, directly through genetic and epigenetic dysregulations and/or indirectly through production of reactive oxygen species (ROS). The purpose of this review is to summarize key findings on the molecular mechanisms altered in cardiac development during exposure to hyperglycemic conditions in utero. It also presents the various in vivo and in vitro techniques used to experimentally model pregestational diabetes. Finally, new approaches are suggested to broaden our understanding of the subject and develop new prevention strategies.

## 1. Introduction

Diabetes mellitus (DM) is a chronic metabolic disorder characterized by elevated blood sugar levels due to impaired insulin secretion or action, or both. The chronic hyperglycemia of DM is associated with long-term damage; dysfunction; and failure of various organs, especially the eyes, kidneys, nerves, heart, and blood vessels. Over the past decades, DM incidence has increased dramatically. In 2021, 537 million people worldwide (age between 20 and 79) were diagnosed with DM, or approximately 1 out of 10 adults. It is estimated that the number of sufferers will reach 643 million by 2030 (International Diabetes Federation 2022). Accordingly, 129.4 million women of childbearing age (20–49 years) are affected by DM, and 20.9 million live births worldwide are at risk of maternal hyperglycemia [1]. The heterogeneity of DM can be explained by its multifactorial etiology; Three different types of DM have been documented: (a) type 1 diabetes (T1D) is an autoimmune disease leading to the destruction of pancreatic β-cells and characterized by insufficient insulin secretion; (b) type 2 diabetes (T2D) is frequently associated with obesity and is characterized by insulin resistance, with the body being unable to use the secreted insulin; (c) gestational diabetes mellitus (GDM) is a reversible diabetic condition that occurs transiently during the second trimester of gestation; and (d) monogenic diabetes, a rare condition resulting from mutations in a single gene [2]. All forms of DM are associated with a high risk of congenital malformations. Diabetic embryopathies affecting the cardiovascular and nervous system can be observed in the offspring of women with pregestational T1D or T2D [3]. Pregestational diabetes causes structural and morphological abnormalities in the developing embryonic heart, whereas GDM induces functional disorders in the fetal heart [4]. Congenital heart defects (CHDs) are the most prevalent type of birth defect. They occur in 9 cases out of 1000 live births and are more frequent in cases of miscarriages [5]. The prevalence of CHDs is increasing worldwide, most probably with enhanced incidence of metabolic disorders including diabetes and obesity [6]. Significant advances in surgery, intervention, and clinical intensive care have resulted in substantial improved survival rates for CHD patients. Subsequently, there is a growing population of adults who have survived CHDs after medical intervention. An important number of CHD survivors suffer from long-term cardiovascular complications and high rates of associated comorbidity. All of the above reasons lead to a substantial increase in healthcare expenditure [7], underscoring the importance of understanding the etiology of CHDs. Advances in genetic testing have identified several genetic factors associated with CHDs. However, the genetic origins of CHDs have been found in only 10% of cases, suggesting an important contribution of environmental modifiers to the etiology of CHDs [8]. Pregestational DM is a non-genetic factor strongly correlated with a high risk of CHDs. The risk associated with gestational diabetes is lower, suggesting that the teratogenicity of glucose is much more severe in the offspring when it occurs early in gestation [9]. Whatever the type of pregestational diabetes (type 1 or 2), the risk of developing a heart defect is identical, i.e., eight-fold higher than in a normal pregnancy [9]. Nevertheless, the molecular mechanisms underlying the teratogenicity of maternal pre-existing DM in the developing heart are still not fully characterized. This review introduces the normal development of the heart, the most common forms of CHD, and the genetic and environmental CHD risk factors. Emphasis is placed on advances in CHD research related to pregestational diabetes. First, the contribution of glucose to normal cardiac development is described; then, the identified molecular mechanisms underlying the teratogenicity of hyperglycemia in cardiogenesis are discussed.

## 2. Congenital Heart Defects

CHDs are structural or functional abnormalities of the heart occurring in utero and detectable during pregnancy or at birth. Cardiac development is a complex process, and perturbations taking place at any level of this process can lead to CHDs. The severity of the CHD phenotype depends on the timing and type of dysregulation. CHDs can range from mild, with little or no effect on the patient, to severe, requiring immediate surgical intervention and leading to mortality and morbidity. There are 20 types of CHDs, which can be subdivided into several categories depending on their spatio-temporal origin during embryogenesis [10,11]. The most common CHD types are summarized in Figure 1.

Firstly, there are conotruncal defects due to a wedging anomaly, i.e., an embedding anomaly of the left part of the ejection segment between the two atrioventricular valves, essential for completion of septation. These include (1) Transposition of the Great Arteries (TGA) [12], in which the pulmonary trunk and the aorta are inverted and connected to the left and right ventricle, respectively; (2) Double-Outlet Right Ventricle (DORV) [13], in which the two major arteries, the pulmonary artery and the aorta, are connected to the right ventricle with Ventricular Septal Defects (VSDs), i.e., when the ventricular septum fails to form correctly, commonly observed with DORV; (3) Persistent Truncus Arteriosus (PTA), also known as Common Arterial Trunk (CAT) [14], when the aorta and pulmonary artery fail to separate into two distinct structures, so the Common Arterial Trunk aligns over a large VSD. (4) Tetralogy of Fallot (TOF) [14] is the combination of several malformations: an overriding aorta that induces VSDs, pulmonary artery stenosis, and right-ventricle hypertrophy. Pulmonary atresia with VSDs is an extreme form of TOF.

Secondly, there are septation defects found isolated or associated to other forms of CHDs. These include (1) the above-mentioned VSDs [14]; (2) Atrial Septal Defects (ASDs) [15], when the septum between the two atria fails to form correctly; and (3) AtrioVentricular Septal Defects (AVSDs) [16], when the four chambers of the heart communicate due to failure of the formation of the atrioventricular septum.

Thirdly, there are left-side obstructive lesions, which occur due to an abnormally formed aorta. These lesions include (1) Hypoplastic Left Heart Syndrome (HLHS) [17], where the left ventricle has no outlet and is thus underdeveloped; (2) Bicuspid Aortic Valve (BAV) [18], the most common type of CHD, where the aortic valve has two leaflets instead of three. (3) Coarctation of the Aorta is a localized narrowing of the aortic lumen that leads to hypertension in the upper limbs; hypertrophy of the left ventricle, if severe; and poor vascularization of the abdominal organs and lower limbs [19].

## 3. CHD Risk Factors

The complexity in understanding CHDs is heightened by their multifactorial etiology. State-of-the-art genetic testing technologies have made it possible to identify genes correlated with a higher risk of CHDs. Genetic testing has been carried out on CHD patients, including those with non-syndromic or isolated CHDs and those with syndromic CHDs [20]. Chromosomal abnormalities, specifically aneuploidy, were one of the first genetic factors associated with CHDs. These abnormalities are found in patients with Down’s syndrome (trisomy 21), 22q11 deletion, and trisomy 18 [21]. Using whole-exome sequencing (WES), single-gene or point mutations were identified in CHD patients. The first mutations identified as responsible for inherited CHDs were found in a group of genes coding for cardiac-specific Transcription Factores (TFs), such as NK2 homeobox 5 (*NKX2.5*), GATA-Binding Protein 4 (*GATA4*), T-box 5 (*TBX5*), and T-box 1 (*TBX1*) [22]. In addition to these genes encoding transcriptional regulators, other genes implicated in cell signaling and laterality pathways have been identified in inherited CHDs (NOTCH1, jagged1 (JAG1) of the Notch pathway, and Nodal and Lefty of the laterality pathway) [23,24]. Moreover, larger-scale whole-genome sequencing made it possible to identify coding and non-coding genomic regions associated with CHDs. Inherited and de novo copy number variations (CNVs) have been associated with CHDs. Some well-known CNVs found in characterized clinical syndromes are del22q11, known as Digeorge syndrome and including a *TBX1* haploinsufficiency [25]; del8p23, including TF *GATA4* haploinsufficiency [26], etc. Additionally, Genome-Wide Association Studies (GWASs) on CHD patients have identified single-nucleotide polymorphisms (SNPs) associated with CHDs [27]. The challenging part with the non-coding genome is to determine the chromosomal loci and the associated genes regulated by this genomic region. Furthermore, pathological variants in genes encoding chromatin-remodeling enzymes have been found in CHD patients, including Chromodomain-Helicase-DNA-Binding Protein 7 (CHD7), Lysine (K)-Specific Methyltransferase 2D (*KMT2D*), Lysine Acetyltransferase 6A (*KATA6A*), Lysine Acetyltransferase 6B (*KATA6B*), Nuclear Receptor-Binding SET-Domain Protein (*NSD1*), Chromodomain-Helicase-DNA-Binding Protein 4 (*CHD4*) [28], suggesting a crucial role of epigenetic regulations in normal heart development.

Each of the aforementioned genetic risk factors is found in barely 10% of all CHD cases, suggesting a multifactorial etiology of CHDs [8]. The in utero environment is a crucial element in the regulation of embryogenesis. Several environmental teratogens have been described in the literature as potential CHD risk factors. These non-inherited CHD-inducing factors include the following: (1) Drugs, notably (a) Thalidomide, a drug used in the 1950s to treat morning sickness during pregnancy and withdrawn from the market in the 1960s because it was found to induce congenital malformations in infants, including CHDs [29]; (b) Isotretinoin, a synthetic form of retinoic acid (RA) used for the treatment of cystic acne, correlated with a higher CHD risk due to dysregulation of RA signaling, which plays a key role in heart morphogenesis [30]; (c) antidepressants: the use of lithium as an antidepressant has been shown to induce an increased risk of CHDs, hypothetically through aberrant induction of Wnt signaling [31,32]. (2) Viral infections: Maternal rubella infection is associated with CHDs, and this is thought to be an indirect effect through induction of hyperthermia [33,34]. (3) Maternal exposure to teratogenic agents, notably smoking [35,36] and alcohol consumption [37,38]. (4) Maternal malnutrition and metabolic disorders: (a) Folic acid or vitamin B9 deficiency has been associated with neurodevelopmental diseases, and more recent studies show that it may also be associated with a higher risk of CHDs [39]; (b) pregestational, type 1, and type 2 diabetes mellitus [40,41]; (c) obesity, which comes along with other health complications, such as type 2 diabetes, is also considered a CHD risk factor, and the isolated effect of obesity on heart development is still unknown [42,43].

## 4. Morphogenesis of the Normal Heart

Understanding the normal heart development is preliminary for investigating CHD etiology. The heart is the first functional organ to develop in the embryo. Cardiac development is a complex process that initiates on embryonic day (E) 7.5 in mice or in the second week of development in humans and culminates in an almost fully formed four-chambered heart by E14.5 in mice or the 8th week of development in humans [44]. Cardiogenesis has been extensively described [44,45,46]. It begins at the end of the gastrula stage, when the embryo separates into the three distinct germ layers: endoderm, mesoderm, and ectoderm. A subtype of progenitor cells originating from the lateral plate mesoderm acquires a precardiogenic potential while migrating anteriorly to gain the midline of the early embryo. These myocardial progenitors are termed First Heart Field (FHF) (Figure 2). FHF cells are known to contribute to the left ventricle, in part to the atria, and other cardiac structures except for the Outflow Tract (OFT). Subsequently, a distinct subtype of myocardial progenitors, known as Second Heart Field (SHF) and originating from the lateral plate splanchnic mesoderm, is added posteriorly to the FHF to form the cardiac crescent, which will fuse into a primary beating heart tube (Figure 2). The SHF is subdivided into anterior SHF and posterior SHF. The anterior SHF is thought to contribute to the OFT and the right ventricle, whereas the posterior SHF is thought to contribute to the InFlow Tract (IFT), the atria, and in part to the OFT. Another subset of endothelial progenitor cells from the same embryonic region, the splanchnic mesoderm, line up in the center of the heart tube to form the endocardium. In this stage, the primary heart tube comprises an outer layer of cardiomyocytes and an inner layer of endocardial cells, both separated by an extracellular matrix called cardiac jelly. A small subset of endocardial cells undergoes Epithelial-to-Mesenchymal Transition (EMT) to form the cardiac cushions, which contribute to the formation of the valves, the membranous interventricular septum, and the atrial septum. Additionally, other progenitor cells not of mesodermal origin but derived from the neuroectoderm also play a role in heart formation (Figure 2). These cells are the Neural Crest Cells (NCCs), which contribute to the OFT, the AtrioVentricular Septum (AVS), and the atrioventricular valves. The linear heart tube is segmented into compartments corresponding to the different structures of the mature four-chambered heart. In each compartment, and following the antero-posterior axis, progenitor cells are predestined to follow an OFT, ventricular, atrial, or IFT cell fate. Once this primary tube has been established, newly differentiated cardiomyocytes are added to the arterial and the venous poles of the heart, leading to elongation and eventual folding, also known as the rightward looping of the heart, regulated by asymmetry-inducing morphogens (Figure 2). The S-shaped heart loop undergoes a series of differentiation–proliferation cycles to balloon and form the heart chambers. In this stage, the right ventricle, therefore, does not have an inlet segment, and the left ventricle does not have an outflow tract. A series of movements then occurs: the conotruncus moves to the left; the atrioventricular canal, to the right; and the interventricular septum, due to the faster growth of the right ventricle, to the left. This series of movements leads to the convergence stage, where the conotruncus, the atrioventricular canal, and the primitive interventricular septum are aligned along the same sagittal plane. This step is absolutely essential to the remodeling of the internal curvature, which leads to the last step, wedging. Wedging is defined as the embedding of the aortic valve, which, through a rotational movement in the counterclockwise direction behind the origin of the pulmonary trunk, comes to be placed between the tricuspid valve and the mitral valve. Afterwards, maturation of the heart continues through the compaction of the myocardial wall, and the development of the atrioventricular septum and the great blood vessels of the systemic and pulmonary blood systems [44,45,47,48] (Figure 2).

Concomitantly, the heart’s conduction system begins to form in early embryonic stages (4th week of gestation in humans or around E8.5 in mice). The cardiac conduction system is responsible for coordinating and regulating the rhythmic contraction of the cardiac chambers, ensuring efficient blood circulation throughout the body. First, the myocardial cells of the primary heart tube can depolarize spontaneously and propagate the electrical impulse; however, their conduction is slow, and their contraction capacity, poor. Subsequently, the SinoAtrial Node (SAN) begins to develop in the venous pole of the embryonic heart, in the IFT region, and serves as the primary cardiac pacemaker. With the looping of the heart, the SAN becomes morphologically distinct and is localized at the junction between the right atrium and the superior caval vein. As development proceeds, the AtrioVentricular Node (AVN), atrioventricular bundle (bundle of His), and Purkinje fibers emerge, permitting coordinated propagation of the electrical impulse within the heart. Understanding the development of the cardiac conduction system is important, as disruptions during this complex process can lead to congenital heart conduction disorders and arrhythmias [49].

## 5. The Role of Glucose Metabolism in Normal Heart Development

The metabolic characteristics of cells vary accordingly to their state of differentiation. During early embryogenesis, glucose is the main source of energy [50]. To understand how glucose provides energy to cells, it is necessary to elucidate intracellular glucose metabolism. In cells, glucose is phosphorylated into glucose-6-phosphate (G6P) by the hexosamine enzyme [51]. G6P is then converted to pyruvate through anaerobic glycolysis, which generates adenosine triphosphate (ATP) (Figure 3). In the mitochondria, pyruvate enters the Krebs cycle and, under aerobic conditions, undergoes oxidative phosphorylation (OXPHOS) to produce large amounts of ATP [52,53,54]. The energy produced through glucose catabolism is essential for cell proliferation and differentiation [55]. Mitochondrial maturation occurs simultaneously with cardiomyocyte maturation in the late-embryonic, fetal, and neonatal stages, preparing the heart for greater energy requirements [55]. This maturation is accompanied by a switch in the energy substrate of cardiomyocytes, with fatty acid metabolism becoming the main source of cellular energy (ATP) [56]. This metabolic shift is due to changes in the expression of genes encoding metabolic enzymes and transporters and acts as a key determinant in driving the cardiac differentiation program [57]. Several glucose transporter proteins are responsible for the facilitated diffusion of glucose from the maternal to the fetal circulation via the placenta. Glucose transporter 1 (GLUT1), an insulin-independent glucose transporter encoded by the Solute Carrier Family 1 Member 2 (*Slc1a2*) gene, is considered the major transporter isoform in early embryogenesis. It is constitutively expressed in the placenta and embryonic cardiomyocytes and is an essential determinant of glucose supply to the embryo [58]. Throughout development, *GLUT1* is gradually downregulated with concomitant upregulation of *GLUT4*, which becomes predominant by the end of gestation. Glucose transporter 4 (GLUT4), an insulin-dependent glucose transporter, needs insulin stimulation to be translocated to the cell membrane, thereby supplying less glucose to the cells [59,60]. This glucose deprivation in late-embryonic/fetal stages is crucial for cardiomyocyte maturation [57].

## 6. Hyperglycemia Teratogenicity during Heart Development

DM usually comes along with other maternal complications. Diabetes is often accompanied by dysregulation of carbohydrates, arachidonic acid, prostaglandins, and inositol metabolisms, making it a syndromic metabolic disorder [61,62]. Together with associated vasculopathies, and reduced or inadequate insulin secretion, these complications make it more difficult to dissect the pathophysiology of CHDs associated with pregestational diabetes. However, the increased maternal blood glucose is considered the primary teratogenic agent in diabetic pregnancies. Previous studies in animal models have shown that even in the absence of DM, hyperglycemia alone can increase the risk of embryopathies [63,64]. Even a transient increase in glucose levels during early gestation (equivalent to the first eight weeks of gestation in humans), also known as the critical period of embryonic development, resulted in higher rates of congenital malformations in the offspring [65]. However, it is unclear whether the duration, timing, and intensity of prenatal exposure to maternal hyperglycemia influence the risk and severity of CHDs in the offspring. The molecular mechanisms underlying the teratogenic effect of excess maternal glucose are the subject of many research studies. As previously mentioned, glucose is used by embryonic cells for fuel via OXPHOS, which generates reactive oxygen species (ROS) (Figure 3). Experimental models of maternal hyperglycemia show upregulation of GLUT1 expression in the embryonic heart, enabling excessive glucose uptake by cardiac cells [66]. Consequently, excessive glucose levels lead to increased ROS production. If the cell’s detoxification mechanisms are insufficient to restore cellular homeostasis, further cellular stress and damage result [67], hence the importance of maintaining strict glycemic control during embryogenesis, to reduce and prevent CHDs. A variety of cardiac malformations have been observed in the offspring of women with pregestational DM. Several meta-analyses conducted by different research groups have consistently identified an increased prevalence of specific CHD phenotypes upon exposure to maternal hyperglycemia. These phenotypes include TGA, heterotaxia, VSDs, and ASDs [41,68]. While T1D and T2D share similar incidence of CHDs, T1D is associated with a higher risk of conotruncal defects and AVSDs, whereas T2D is more commonly associated with heterotaxia and left-ventricular OFT obstructive abnormalities [69]. Congenital anomalies of the heart have been observed following exposure to high glucose levels in both pregestational and gestational diabetes. However, gestational diabetes mellitus is only weakly associated with structural defects in offspring. Population-based studies on offspring of pregestational diabetic mothers showed that CHDs originating from an altered differentiation of the anterior SHF progenitor cells (PTA, TOF, DORV, HLHS, and VSDs) (Figure 1 and Figure 2) were more frequent than other CHD types [9]. This suggests that progenitors of the anterior SHF are more sensitive to maternal hyperglycemia than those of the posterior SHF [70,71]. Nonetheless, ASD cases in diabetic pregnancies were also reported [9]. Although less prevalent, these ASDs suggest that maternal hyperglycemia impacts the patterning of the posterior SHF [68,70].

The progenitors of the second heart field have been shown to contribute to both the heart and skeletal muscles of the head and neck. They give rise to not only cardiomyocytes but also endothelial, endocardial, and smooth/skeletal muscle cells [72,73,74]. Since hyperglycemia during pregnancy is associated with various congenital defects, exploring the association between a CHD type and congenital defects affecting the head and neck muscle lineages in patients could yield significant insights. Another significant aspect along these lines is that CHDs are frequently accompanied by genetic syndromes presenting both cardiac and extra-cardiac anomalies [72,75,76]. DiGeorge syndrome (22q11.2 deletion) is a striking example, as it is associated with prevalent cardiac malformations and craniofacial defects. Conotruncal lesions such as interrupted aortic arch, truncus arteriosus, tetralogy of Fallot, and ventricular septal defects are frequently diagnosed in children with 22q11 deletion. The gene *TBX1*, located on chromosome 22q11.21, has been found in patients with the DiGeorge syndrome. Since *Tbx1* is expressed in the second heart field progenitor cells and is altered by maternal diabetes [77], the prevalence of TBX1-associated CHDs may correlate with craniofacial defects upon exposure to maternal hyperglycemia. To date, there have been no documented reports on this association in either human clinical data or mouse studies.

In addition to structural and morphological changes, pregestational and gestational diabetes can also lead to functional disorders. These functional changes include cardiac hypertrophy, increased heart rate, impaired ventricular filling, and outflow tract obstruction, resulting in systolic and diastolic dysfunction and decreased overall myocardial performance [78,79,80]. Although cardiac hypertrophy usually disappears spontaneously after birth, there is growing concern that cardiac dysfunction persists into early childhood, with possible long-term effects that may lead to future cardiovascular disease [81]. Cardiac hypertrophy has been found to be linked to fetal hyperinsulinemia and insulin-like growth factor I (IGF-1) deregulation. IGF-1 promotes cardiomyocyte hypertrophy, leading to functional impairment [82,83].

Monitoring fetal cardiac rhythm using electrocardiogram (ECG) showed that hyperglycemia in mothers with T1D induced an accelerated fetal heart rate [84]. Fetal heart rate abnormalities occur in 1-to-2% of pregnancies and may be associated to structural abnormalities of the heart [85]. Further studies are needed to identify the cellular and molecular mechanisms behind fetal dysrhythmias in response to hyperglycemia.

## 7. Modeling Pregestational Diabetes

Pre-existing diabetes unquestionably exerts an adverse impact on both pregnancy and the cardiac development of the fetus, even when women maintain adequate glycemic control. With the growing number of women affected by this condition in recent times, it is crucial to adopt a proactive approach when it comes to educating, preventing, and managing the metabolic health of these patients. There is a crucial need to pursue research efforts in this domain to gain comprehensive insights into the multifaceted connection between maternal diabetes and fetal cardiac abnormalities. Additionally, genetic factors may amplify susceptibility to specific environmental influences, a phenomenon that necessitates clarification. There are ongoing investigations into potential treatments for pregnant women with diabetes. Future studies using complementary experimental models should assess whether these potential treatments effectively reduce the risk of CHDs in offspring exposed to them. These experimental models could help pinpoint essential regulatory checkpoints throughout multiple stages of cardiac development, leading to uncovering why infants exposed to teratogenic agents like hyperglycemia are more susceptible to fetal cardiac developmental issues. By utilizing dynamic methodologies, single-cell transcriptomics, epigenetic investigations, and lineage analysis, we could improve our comprehension of how maternal hyperglycemia affects these stages and how potential treatments may reduce the risk of CHDs in offspring. This would facilitate early and efficient interventions as well as the development of effective therapies.

(a)

*in vivo*

 models


To better understand the high risk of CHDs in response to placental transfer of elevated maternal blood glucose to the developing embryo, researchers are relying on mammalian models of pregestational diabetes. The similarity in embryonic development between humans and rodents has made the latter a valuable research tool. Having a four-chambered heart and a structure reminiscent of the human heart allows us to study cardiac development and cardiac malformations [86]. The literature describes numerous rodent models of DM used to study the features of pregestational diabetes in cardiac development (Table 1): (a) Chemically induced mouse or rat models: The most commonly used diabetic mouse model is the streptozotocin (STZ)-induced model. This model involves the injection of the pancreatic toxin streptozotocin to specifically destroy insulin-producing β-cells, mimicking T1D with insulin deficiency and hyperglycemia [87,88]. (b) Natural mutations: Rodent models with naturally occurring mutations are another alternative for T1D. This model does not require external induction of diabetes, and it includes the non-obese diabetic (NOD) mouse model [89], the bio-breeding (BB) rat model [90], and the Cohen rat model [91]. Spontaneous autoimmune disease leads to hyperinsulinemia and hyperglycemia through the destruction of pancreatic β-cells [90,91,92]. (c) Diet-induced mouse or rat models: These models consist of feeding animals with a high-fat diet (HFD), leading to obesity and insulin resistance, recapitulating the main features of type 2 diabetes [93]. (d) Genetically modified rodent models of pregestational diabetes: Ins2^Akita/+^ mice, known as Akita mice, have an autosomal dominant heterozygous mutation in the insulin2 gene, preventing the normal folding and cleavage of pro-insulin, leading to its accumulation. This is an established model of T1D with hyperinsulinemic hyperglycemia [57]. Non-mammalian animal models have also been used to study the effect of hyperglycemia on cardiac development. Jaime-Cruz et al. studied the effect of embryonic hyperglycemia on cardiogenesis in fertilized Bovans chicken eggs. Hyperglycemia was induced by injecting glucose for 10 consecutive days through the eggshell window [94]. Hyperglycemia was also shown to alter cardiac looping in zebrafish embryos. Hyperglycemia was induced by treating the embryos, 6 h post-fertilization, with 25 mmoL/L D-glucose or L-glucose for 24 h [95].

Many other animal models are currently studied for diabetes. However, many have not been used to target offspring to explore the risk of CHDs induced by maternal diabetes (e.g., GK rat model for type 2 DM) [96].

(b)
*in vitro* models


Human heart organoids, deriving from human pluripotent stem cells (hPSCs), have recently been used to model pregestational DM and study the associated effect on cardiac development [97] (Table 1). To mimic maternal DM, human organoids were cultured under conditions recapitulating the main feature of T1D and T2D pregestational diabetes (11.1 mM glucose and 1.14 nM insulin). In contrast, control human heart organoids were cultured under physiological glucose and insulin conditions reported in normal mothers (3.5 mM glucose, 170 pM insulin) [97]. Moreover, to study the effect of maternal hyperglycemia on endocardial cells, an immortalized embryonic cardiac cell line of atrioventricular cushion mesenchymal cells was used [98]. These cells were treated with 25 mM D-glucose under hyperglycemic conditions versus 5.5 mM under normoglycemic conditions [98]. To study the effect of hyperglycemia on cardiomyocyte differentiation and maturation, human embryonic stem cells (hESCs) were differentiated into cardiomyocytes in reduced versus excessive levels of glucose [57,99].

## 8. Molecular Mechanisms Underlying the Teratogenic Effect of Maternal Pregestational Diabetes

Glucose might exert a teratogenic effect on heart development via a signaling pathway regulating insulin responsiveness. Insulin sensitivity is indeed involved in the physiopathology of T1D and T2D. Moreover, insulin and associated signaling pathways are critical regulators of embryogenesis and early development through the regulation of proliferation and differentiation [100]. Transcriptomic studies have shown that many genes required for normal embryonic heart development are dysregulated due to in utero exposure to a hyperglycemic environment (Table 2). Most of these genes encode for transcription factors, DNA-binding proteins, effectors of molecular pathways, cell surface receptors, extracellular matrix adhesion molecules, chromatin remodelers, etc. Essential biological functions can be impacted by dysregulated genes, including the remodeling of the extracellular matrix and cell cytoskeleton, cell metabolism, differentiation, proliferation, and apoptosis [87,101,102].

(a)
Oxidative stress, endoplasmic reticulum stress, and apoptosis


Cells require oxygen to breathe. The reduction of oxygen produces free radicals and hydrogen peroxide, highly reactive molecules known as reactive oxygen species (ROS). The produced ROS play a major role in cell signaling and physiology, including proliferation, differentiation, and migration [122]. Impaired redox balance favoring excessive ROS production, together with deficient cellular antioxidant defense mechanisms, is referred to as oxidative stress (OS). OS induces cellular damage and death through the oxidation of proteins, lipids, and DNA [122]. It has been shown to be a key mechanism underlying the teratogenicity associated with diabetic pregnancy, causing developmental delays, neural tube defects, and cardiovascular abnormalities [62,123]. OS is thought to disrupt cellular homeostasis and increase the risk of CHDs in the offspring of diabetic mothers through the altering of key molecular pathways required for normal heart development (Transforming Growth Factor-β (TGFβ) and Wingless (Wnt) pathways) and the inducing of excessive apoptosis [113,114,115,124,125,126]. Numerous studies have shown that maternal high blood glucose levels correlate with higher ROS production and dysregulation of antioxidant mechanisms in many embryonic tissues, including embryonic hearts. In rat models of pregestational DM, significantly reduced levels of antioxidant enzymes, including superoxide dismutase (SOD1), catalase, and glutathione peroxidase, were observed in embryonic and fetal tissues [121,127]. Other studies have shown that antioxidant treatment can reverse the deleterious effect of hyperglycemia and reduce embryopathies [109]. The incidence of CHDs was significantly reduced in the offspring of pregestational diabetic female mice supplemented with N-acetyl cysteine (NAC) in drinking water [109]. NAC supplementation increased cell proliferation, reduced apoptosis, and restored the expression of key cardiac development genes in fetal heart cells [109]. Supplementation with zinc, an essential cofactor for scavenger-enzyme activity, reduced the incidence of CHDs and maternal hyperglycemia-associated oxidative stress in embryonic mouse hearts [123]. Moreover, overexpression of the SOD-1 transgene reduced the incidence of CHDs and apoptosis in fetal hearts exposed to pre-existing maternal DM through the restoration of impaired Wnt and TGFβ signaling [114,115].

Whether the teratogenic effect of hyperglycemia in diabetic pregnancies acts through the direct alteration in gene expression or indirectly through associated oxidative stress and hypoxia processes remains controversial. Wang et al. showed that the Apoptosis signal-regulated kinase 1 (*Ask1*) encoding gene, activated by oxidative stress in diabetic pregnancies, exerts a teratogenic effect on cardiac development through the induction of ER stress and apoptosis, and the impairment of essential cardiogenic factors. Moreover, deletion of the *Ask1* encoding gene reduced the incidence of septation and OFT defects and restored the levels of *Bmp4, NKX2.5*, and *Gata4* in embryonic mouse hearts exposed to maternal hyperglycemia [113]. Another recent study in STZ-induced rat models of pregestational diabetes showed that maternal hyperglycemia induced an increase in the activity of Forkhead Box O1 (*Foxo1*) and, subsequently, its downstream effector genes in embryonic cardiac explants. These genes are pro-oxidant and pro-inflammatory, leading to oxidative stress, inflammation, and a high risk of defective cardiogenesis [128]. Maternal hyperglycemia has also been shown to induce Endoplasmic Reticulum (ER) stress. Markers of ER stress, such as phosphorylated protein kinase R (PKR)-like endoplasmic reticulum kinase (p-PERK), phosphorylated Eukaryotic translation initiation factor 2A (p-eIF2α), phosphorylated Inositol-requiring enzyme 1 (p-IRE1α), immunoglobulin heavy chain-binding protein (BiP), and C/EBP Homologous Protein (CHOP), were found to be upregulated in mouse embryonic hearts exposed to maternal hyperglycemia compared with those exposed to control glycemic conditions [113,116]. A recent study on human cardiac organoids showed that PGD induced ER stress and dysregulation of lipid metabolism. Treatment of ER stress reduced the deleterious effects associated with PGD in human cardiac organoids [129].

(b)
Dysregulated signaling pathways


Maternal hyperglycemia disrupts cardiogenesis by altering key pathways regulating normal heart development. The Notch pathway, activated in endocardial cells, induces the expression of VE-cadherin transcriptional repressors, leading to endocardial-cell EMT to form cardiac cushions [130,131]. In addition, Notch signaling regulates the expression of Nodal-related genes involved in the morphogenesis of left–right asymmetry [132]. Hyperglycemia has been shown to disrupt Notch signaling in the early stages of embryonic development (embryonic day E6-7.5), thus affecting the establishment of the left–right axis in mouse embryos [132]. In addition, endothelial Nitric Oxide Synthase (eNOS), an enzyme specifically expressed in endothelial and myocardial cells, provides nitric oxide (NO) to cells. eNOS signaling and NO signaling play a crucial role in cardiac development, and null mutations of eNOS are associated with a high CHD risk [133]. Decreased NO levels and the uncoupling of eNOS activity have been reported in the fetal hearts of diabetic female mice [107]. Treatment of diabetic female mice with sapropterin, an orally administered synthetic form of eNOS cofactor, recoupled eNOS; reduced ROS production; and subsequently rested the expression of cardiac transcription factors such as Gata4, Gata5, Nkx2.5, Tbx5, and Bmp10 [105]. These data suggest that the normalization of eNOS activity through Sapropterin administration could be a potential therapeutic approach to preventing CHDs associated with pregestational diabetes. Furthermore, NO is known to interact with Notch signaling during embryonic heart development. Under hyperglycemic conditions, decreased NO levels in cultured cushion mesenchymal cells, in response to eNOS dysfunction, have been associated with increased levels of Jarid2, part of the repressive polycomb 2 complex, which suppresses Notch1 expression in these cells [98]. Another important signaling pathway is the canonical and non-canonical Wnt pathway. It regulates cardiac NCCs, and the deletion of key effectors in this pathway induced OFT defects similar to those observed in diabetic pregnancy-associated cardiopathies [134]. In a study by Wang et al., the authors found that two Wnt antagonists, secreted frizzled-related protein 1 (sRfp1) and Dickkopf Wnt signaling pathway inhibitor 1 (Dkk1), were overexpressed in the embryonic hearts of streptozotocin (STZ)-induced diabetic female mice. In addition, they observed decreased phosphorylation of Dishvelled protein (Dvl2), increased Glycogen Synthase Kinase 3 Beta (Gsk3β) activity of canonical Wnt signaling, and decreased Wnt5a expression of non-canonical Wnt signaling [114]. Another study by Zhao et al. showed that endogenous inhibitors of Wnt signaling were upregulated in the hearts of E10.5 embryos exposed to maternal hyperglycemia in STZ-induced mouse models, including Wnt inhibitory factor 1 (Wif1), amino-terminal enhancer of split (Aes), and the β-catenin destructor Casein kinase 2 a1 polypeptide (Csnk2a1) [117]. All of the above studies suggest the association of cushion and OFT defects in hyperglycemic pregnancies with suppression of Wnt signaling. Another important signaling pathway whose repression has been demonstrated in the heart of embryos exposed to maternal hyperglycemia is the TGFβ pathway. In normal heart development, TGFβ signaling has a crucial function in cardiac NCC migration, homing, and maturation. Mutations in one or more components of this pathway are known to contribute to OFT and septal defects [135]. Zhao et al. showed that several genes of the TGFβ signaling analyzed using real-time RT-PCR were differentially expressed upon exposure to maternal hyperglycemia, including the TGFβ3 ligand and the TβR1 and TβR2 receptors [118]. Accordingly, Wang et al. showed that TGFβ ligands (TGFβ1 and TGFβ3) were downregulated in E12.5 mouse embryonic hearts of STZ-induced diabetic mothers. A reduction in phosphorylation levels of TGFβ downstream effectors, such as pSmad2/3, and in the expression of TGFβ-regulated genes, such as Snail Family Transcriptional Repressor 2 (Snai2), Connective tissue growth factor (Ctgf), Growth differentiation factor 1 (Gdf1), were also reported. Overexpression of SOD1 in transgenic embryos was able to revert the molecular dysregulations of TGFβ signaling induced by maternal hyperglycemia, but the effect of SOD1 rescue on the cardiac phenotype has not been demonstrated [115]. These data suggest an interaction between oxidative stress and the TGFβ pathway in heart development. Another signaling pathway has been associated with the teratogenic effect of pregestational maternal diabetes: the Hypoxia-Inducible Factor-1 (Hif1) pathway. Hif1 is a cellular oxygen sensor and, more recently, has been shown to regulate glycolytic genes. In E10.5 mouse embryos exposed to maternal hyperglycemia, the level of Hif1α protein was upregulated, and 20 out of 22 Hif1α-responsive genes were upregulated, including enzymes regulating glucose metabolism, such as Glut1 [101].

(c)
Dysregulated transcription factors encoding gene expression


Genes required for normal cardiac morphogenesis are transcriptionally regulated by several key TFs. Any alteration in TF functionality affects the expression of its downstream genes, leading to disruption of the cell transcriptional profile and more or less biased cellular physiology. Microarray genechip analysis revealed that several genes coding for TFs were significantly altered in embryonic/fetal heart tissue from mice from diabetic mothers [102]. The TF-encoding genes altered by maternal diabetes are genes associated with CHDs, such as *Nkx2.5*, *Gata4*, *Tbx5*, *Tbx1*, etc. Other studies have targeted more specific genes, previously described in the literature as essential factors in normal cardiogenesis, and have investigated the effect of maternal hyperglycemia on the expression and function of these genes, as well as their contribution to CHDs. Paired box 3 (*Pax3*), whose function is to ensure NCC migration and OFT septation, was also downregulated by maternal DM in the NCCs of embryonic hearts [120]. This study showed that pre-existing maternal DM leads to impaired OFT formation, resulting in conotruncal heart defects. Recent advances in single-cell biology have made it possible to study the effect of maternal hyperglycemia on different cardiac cell subtypes [120]. To investigate the effect of maternal hyperglycemia on early cardiac development, Manivannan et al. performed single-cell RNA sequencing (scRNA-seq) on embryonic hearts from E9.5 and E11.5 mice exposed to hyperglycemic (STZ-induced diabetic mouse model) and normoglycemic conditions *in utero*. They found transcriptomic changes in a subtype of SHF multipotent progenitors upon exposure to maternal hyperglycemia. In these Islet 1-positive (Isl1+) multipotent progenitor cells, maternal hyperglycemia induced differential expression of genes encoding TFs regulating cell lineage specification (Insulin Gene Enhancer Protein (*Isl1*), *Tbx1*, *Tbx20*, Fibroblast growth factor 10 (*Fgf10*), Myocyte enhancer factor 2C (*Mef2c*), *Nkx2.5*, and Heart And Neural Crest Derivatives Expressed 2 (*Hand2*)), thus contributing to CHDs [77].

(d)
Dysregulated epigenetic regulation


The maternal in utero environment plays a crucial role in embryonic heart development. When this environment is compromised or unhealthy, it can potentially lead to birth defects via epigenetic dysregulation. Epigenetic regulations include post-translational modifications of histones and DNA methylation, which modulate chromatin accessibility and allow the transition from a transcriptionally active chromatin state (euchromatin) to an inactive chromatin state [136,137]. Advances in epigenomic approaches, such as Assay for Transposase-Accessible Chromatin coupled to high-throughput sequencing (ATAC-seq), have made it possible to identify *cis*-regulatory elements and map chromatin accessibility in healthy versus unhealthy cellular states. Using ATAC-seq, it was demonstrated that hyperglycemia induces epigenetic changes in cultured mesenchymal cells of the atrioventricular cushion [98]. Excess glucose acts by altering chromatin configuration in the regulatory region of the *Nos3 locus*, resulting in a downregulation of NO and an eventual increase in Jarid2 levels [98]. Jarid2 is a repressor of Notch1, and as previously mentioned, impaired Notch signaling is associated with CHDs [98]. Studies about the impact of maternal hyperglycemia on the epigenome are still limited. Further approaches targeting DNA methylation, histone modification, and chromatin remodeling could provide mechanistic insights into how maternal hyperglycemia alters cardiac differentiation.

## 9. Heightened CHD Risk Due to Combined Effect of Genetic Predisposition and Hyperglycemia

The hypothesis of a multifactorial etiology of CHDs was introduced in 1968, suggesting an interaction between heredity and the in utero environment [138]. Currently, interactions between environmental factors and candidate genes in the contribution to birth defects are being investigated, which explains why experimental evidence on this topic is still limited. Candidate genes associated with a specific birth defect are selected from several sources, including genes identified in cohorts suffering from congenital abnormalities and transcriptionally deregulated genes identified in animal studies of congenital malformations [139]. More specifically, in CHDs, the impact of a gene–environment interaction on cardiac development has been reported in several deleterious maternal conditions. In a context of maternal hypoxia, the combined effect of low-oxygen exposure and haploinsufficiency in genes associated with monogenic CHDs in humans was investigated. At 8% oxygen, there was no significant difference in the incidence and severity of CHDs in *Tbx1^+/−^* or *Nkx2.5^+/−^* fetuses compared with *Tbx1^+/+^* or *Nkx2.5^+/+^* fetuses, respectively [140], whereas at 5.5% oxygen, a significantly higher CHD risk was observed in *Tbx1^+/−^* compared with *Tbx1^+/+^* of the same litter exposed to low oxygen levels. Accordingly, a significant increase in embryonic lethality was noted in *Nkx2.5^+/−^* embryos compared with *Nkx2.5^+/+^* embryos of the same litter upon exposure to low oxygen levels [140]. Given that maternal hyperglycemia is associated with hypoxia, similar molecular dysregulations could occur in diabetic pregnancies when associated with genetic risk factors [141]. These data suggest an accentuated effect following a two-hit phenomenon (genetic predisposition and environmental perturbation) on the penetrance and severity of CHDs. Similarly, more and more research studies are focusing on the combined effect of genetic predisposition to CHDs in maternal pregestational diabetes on the induction of CHDs. A retrospective study demonstrated that maternal diabetes, *NKX2.5* variants, and their interactions were significantly associated with CHDs in offspring [142]. The interaction between these two factors deserves further investigation, especially in relation to the specific subtypes of CHDs involved. When diabetic female mice were crossed with males having a heterozygous-null mutation in the *Hif1α* gene, increased susceptibility to CHDs was observed in *Hif1α^+/−^* embryos exposed to maternal hyperglycemia compared with their WT littermates [104]. Precisely, the combination of *Hif1α* heterozygous-null mutation and maternal DM induced a significant decrease in the number of embryos per litter and an increase in CHD incidence, notably AVSDs. Additionally, it led to differential expression of key cardiogenic TFs, including *Nkx2.5*, *Tbx5*, and *Mef2*c, in hearts of *Hif1α^+/−^* embryos exposed to maternal DM compared with Hif1α^+/+^ littermates [104]. Furthermore, an interaction between maternal diabetes and *Notch1* haploinsufficiency induced a significant increase in VSD incidence in a mouse model of diabetic pregnancy [98]. Genetic susceptibility can be influenced by environmental factors, and conversely, environmental factors can also be influenced by genetic factors [143]. A recent study underscores the necessity of exploring the mechanisms that link CHD risk with modifiable risk factors [144]. The authors’ conclusion is that genome-wide de novo variants are not associated with maternal diabetes or obesity in cases of congenital heart disease. This implies that other factors, such as inherited genetic variants, the metabolic influence of the mother on the developing heart, or environmental factors, should be considered crucial areas for future research to gain a better understanding of their impact on CHD risk. It is worth noting that numerous pathogenic CHD variants display varying penetrance and expressivity, suggesting that environmental factors could potentially modify the severity of CHDs. Consistent with this, mouse models of CHDs have demonstrated that the penetrance of NOTCH1-related CHDs is heightened when exposed to gestational hypoxia [145]. All of the above-mentioned evidence supports the hypothesis of increased susceptibility to CHDs as a result of a gene–environment interaction.

## 10. Combined Teratogenicity of Diabetes and Obesity in CHDs

Maternal obesity (body mass index BMI > 30) is linked to unfavorable pregnancy outcomes and neonatal complications. These adverse effects include stillbirth, macrosomia, and congenital malformations such as neural tube defects and CHDs [146,147,148]. This health concern holds important significance, as both the incidence and severity of maternal obesity has dramatically increased during recent years [149]. Currently, >15% of women of procreation age worldwide suffer from obesity, while 40% are overweight (BMI > 25) (World Health Organization 2018). Maternal obesity is associated with an increased risk of a wide range of CHDs, including septal defects, aortic arch defects, conotruncal defects, and left-ventricular outflow tract obstruction defects [150]. Obesity is usually accompanied by other metabolic disorders, like DM. The prevalence of pregestational DM in obese women ranges from 0.6 to 3.8% [151]. The use of non-obese diabetic animal models enables the study of the isolated effect of hyperglycemia on embryonic development, while excluding the lipotoxic effect of maternal obesity. Maternal pre-existing DM was shown to partially mediate the risk of CHDs associated with maternal obesity [151]. The adjustment of glucose levels in obese pregnant women plays an important role in reducing the risk of CHDs [151]. However, it remains unclear whether the risk of congenital malformations is identical in cases of pre-existing maternal obesity and in cases of weight gain during pregnancy.

## 11. Future Directions and Conclusions

Although the molecular mechanisms underlying the elevated CHD risk in maternal pregestational diabetes are being identified, available data are still limited, and further investigations need to be performed. Numerous studies highlight the teratogenic effect of maternal hyperglycemia through the induction of oxidative stress (Figure 4). This oxidative stress disrupts cellular homeostasis by inducing transcriptional changes targeting cell proliferation, differentiation, migration, apoptosis, and inflammation. Additionally, maternal hyperglycemia alters, directly or indirectly via oxidative stress, important molecular pathways implicated in cardiac development, notably the Wnt, Notch, TGFβ, NO, and Hif1α pathways. Moreover, epigenetic changes associated with maternal hyperglycemia intervene in the cell’s transcriptional program, leading to altered cellular physiology. The emergence of high-throughput genomic techniques could help broaden our understanding of and gain new insights into the molecular etiology of CHDs associated with pregestational diabetes. Single-cell (sc) multi-omics approaches, combining scRNA-Seq and scATAC-Seq, could be extremely useful in decoding the mechanisms underlying these perturbations [152]. This emerging method would enable the screening of hyperglycemia-induced alterations in each of the different subtypes of cardiac progenitor cells and differentiated cardiomyocytes and thus those occurring in the early stages of cardiac morphogenesis. In addition to ATAC, which targets chromatin accessibility, other “on the rise” sc or special epigenomic approaches need to be performed, in order to cover all the layers of the epigenome, including DNA methylation, histone modification, and chromatin architecture [153]. Combining all of the above-mentioned innovative epigenomic methods could help unravel the mechanisms of maternal hyperglycemia teratogenicity and ultimately translate these findings to humans with potential preventions and treatments.

Since 1980, our knowledge of cardiac diseases has evolved according to the concept of developmental origins of health and disease (DOHaD) [154]. DOHaD suggests that exposure to a deleterious environment during the early stages of development (preconception, gestation, and early post-natal stages), significantly impacts the health of individuals during their adult life [154]. In this case, even if children born to mothers with pregestational diabetes appear to have normal heart structure and functionality, this does not exclude the probability of developing cardiovascular diseases (CVDs) in adult life associated with in utero exposure to a hyperglycemic environment. Research studies on animal models have shown that exposure to deleterious environmental conditions predisposes offspring to cardiac diseases through epigenetic signature inheritance [155]. Although maternal nutrition has been the most extensively studied to date, a growing number of studies also attest to the influence of the paternal nutritional environment on the phenotype of the offspring [156] (Figure 4). It would be of great interest to test if paternal DM is also associated with a higher risk of CHDs and CVDs in offspring.

## Figures and Tables

**Figure 1 ijms-24-16258-f001:**
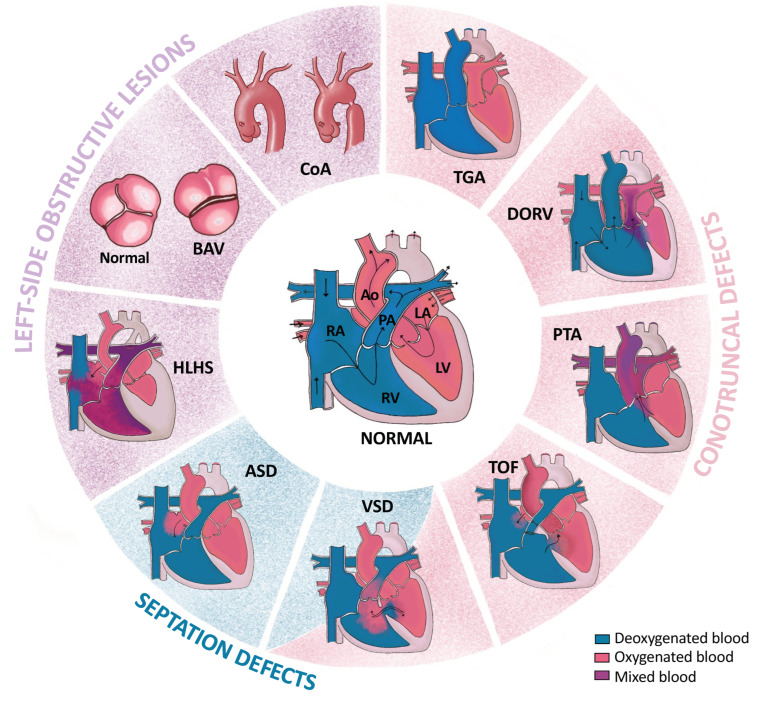
Schematic representation of the normal four-chambered mammalian heart and the structural abnormalities of the most common forms of congenital heart defects (CHDs). Black arrows indicate the direction of blood flow. TGA: Transposition of the Great Arteries; DORV: Double-Outlet Right Ventricle; PTA: Persistent Truncus Arteriosus; TOF: Tetralogy of Fallot; VSD: Ventricular Septal Defect; ASD: Atrial Septal Defect; HLHS: Hypoplastic Left Heart Syndrome; BAV: Bicuspid Aortic Valve; and CoA: Coarctation of the Aorta; RA: Right Atrium; RV: Right Ventricle; LA: Left Atrium; LV: Left Ventricle; PA: Pulmonary Artery; Ao: Aorta.

**Figure 2 ijms-24-16258-f002:**
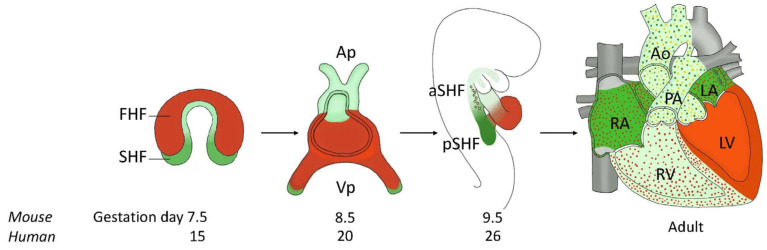
Representative scheme of the main stages of cardiac development. First, on gestation day 15 (equivalent to gestation day 7.5 in mice), the two cardiac lineages derived from the embryonic mesoderm, the First Heart Field (FHF) progenitor cells (red) and the Second Heart Field (SHF) progenitor cells (green), form the cardiac crescent. Around day 20 (equivalent to day 8.5 in mice), the 2 heart fields merge to form the elongated heart tube, which loops rightward on day 26 (equivalent to day 9.5 in mice). The looped heart contains the FHF, which begins to differentiate into left ventricle cardiomyocytes (red), anterior SHF (aSHF, light green), posterior SHF (pSHF, dark green), and migrating cardiac neural crest cells (yellow dots). Cardiogenesis proceeds through a series of proliferation, differentiation, and maturation, culminating in a four-chambered adult heart (D). Ap: Arterial pole; Vp: Venous pole; V: Ventricle; A: Atrium; RA: Right Atrium; RV: Right Ventricle; LA: Left Atrium; LV: Left Ventricle; PA: Pulmonary Artery; Ao: Aorta.

**Figure 3 ijms-24-16258-f003:**
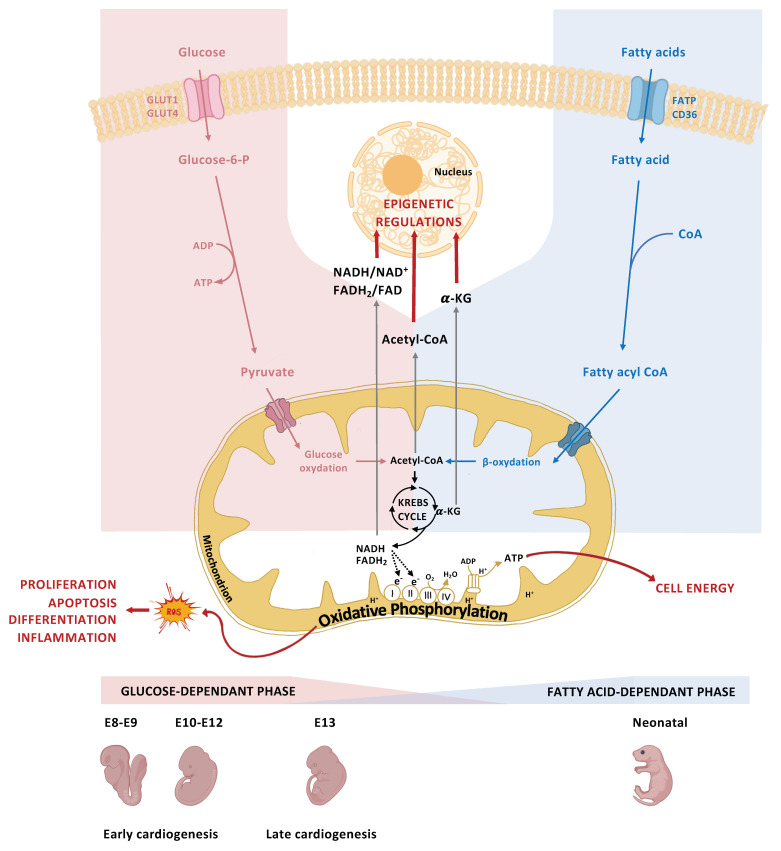
Role of glucose metabolism in cell physiology, genetics, and epigenetics. (**Upper panel**) In the cell, glucose and fatty acid are metabolized into pyruvate and fatty acyl CoA, respectively, which are converted to acetyl-CoA in the mitochondria. In a process known as oxidative phosphorylation (OXPHOS), acetyl-CoA enters the Krebs cycle, generating an energy substrate for the cell (ATP), cofactors, and Reactive Oxygen Species (ROS). The molecules generated, such as NADH and α-KG, serve as cofactors for epigenetic remodeling enzymes. Hyperglycemia leads to excessive ROS production. If the cells defense mechanisms (antioxidant activity) fail to restore homeostasis, oxidative stress results. ROS can induce cellular damage by regulating cell proliferation, differentiation, apoptosis, and inflammation. (**Lower panel**) During early cardiogenesis, glucose is the main source of energy. The transition from embryonic stage to fetal stage (the maturing of the heart) is accompanied by a switch in the energy substrate to fatty acid.

**Figure 4 ijms-24-16258-f004:**
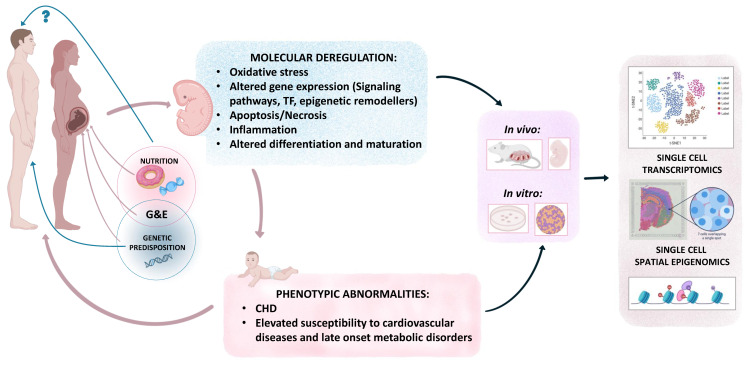
Gene–environment interaction (G&E) as a two-hit risk factor for congenital heart defects associated with pregestational diabetes. This scheme illustrates the interaction between genetic predisposition to CHDs and maternal hyperglycemia in embryonic heart development. The molecular and structural abnormalities of the developing heart resulting from this G&E interaction are shown. In vivo and in vitro models are proposed to examine genomic (transcriptomic and epigenomic) changes in response to maternal hyperglycemia, using single-cell approaches. The contribution of the paternal genetics and nutrition to abnormal heart development is also suggested.

**Table 1 ijms-24-16258-t001:** Advantages and disadvantages of in vivo and in vitro models of pregestational diabetes.

	Diabetes Mellitus Models	Advantages	Disadvantages
** *in vivo* **	**Chemically induced models:** STZ-induced	Quick (rapid onset after STZ injection) and cost-effective method.	Cytotoxicity of STZ to liver and kidneys and mild teratogenic effect.
		Simulate T1D and when combined to HFD can simulate T2D.	Heterogeneous response to STZ (genetic background, and strain).
	**Genetic models:** spontaneous mutations (NOD, BioBreeding and Cohen rats) or transgenic models (Akita mice)	Simulate T1D.	Use of genetic models in cardiac development is limited.
		No chemical toxicity on vital organs.	Time needed to generate transgenic diabetic females (multiple rounds of breeding to get the desired genotype).
			Diabetic models induced by spontaneous mutations can take up to 30 weeks to show diabetic characteristics.
	**Diet-induced:** HFD	Recapitulate key features of T2D (hyperglycemia, insulin resistance and obesity).	Time consuming: it takes 15 weeks of HFD to get a diabetic state (not recommended for developmental biology studies as old mice have reduced fertility).
		No insulin deficiency.	Lipotoxicity making it hard to study the isolated effect of glucotoxicity.
		No chemical toxicity on vital organs.	
** *in vitro* **	**2D culture**	Permit to study the effect of an isolated environmental factor (ex: hyperglycemia and/or hyperinsulenmia) on a specific cardiac cell type.	Need for in vivo validation.
		Large scale screening.	Does not reproduce system biology with the complex interactions between different organs and tissues.
		Quick and easy.	
		Reduce the use of animal models and less ethical issues.	
	**3D culture:** organoïds	Permit to study the effect of an isolated environmental factor (ex: hyperglycemia and/or hyperinsulenmia): simultaneously on several subtypes of cardiac cells, and also at different stages of specification and differentiation.	Need for in vivo validation.
		More physiological context than 2D culture.	Does not reproduce system biology, with the complex interactions between different organs and tissues.
		Large scale screening.	Lack of standardization and reproducibility in generating cardiac organoids.
		Quick and easy.	
		Reduce the use of animal models and less ethical issues.	

**Table 2 ijms-24-16258-t002:** Dysregulated genes implicated in cardiac development upon exposure to a hyperglycemic environment.

Experimental Animal Model of Pregestational Diabetes	Observed CHD	Dysregulated Genes	Experimental Approach	Reference
STZ-induced Sprague Dawley rats	N/A	68 upregulated genes 271 downregulated genes	RNA-Seq	[103]
STZ-induced C57BL/6 mice	VSD	*Notch1, Hey2, EfnB2, Nrg1, Bmp10, Jarid2* and *Nos3*	RT-PCR	[98]
STZ-induced FVB mice	VSD and thin myocardial wall	*Hif1α, Nkx2.5, Tbx5, Mef2C, α-SMA Cx43, Nppa*	RT-PCR	[104]
STZ-induced C57BL/6 mice	ASD, VSD, AVSD, PTA, DORV, pulmonary valve stenosis, aortic valve stenosis, RV hypertrophy, HLHS, narrowing of the aorta	*Gata4, Tbx5, Nkx2.5, Gata5, Bmp10, Notch1, Gch1* and *Dhfr*	RT-PCR	[105]
STZ-induced CD1 mice	N/A	*Bcl2, Casp3,* and *Casp9*	RT-PCR	[106]
STZ-induced Swiss Albino mice	PTA, VSD, AVSD and defective valve development	*Bmp4, Msx1,* and *Pax3*	RT-PCR	[87]
STZ-induced Swiss Albino mice	N/A	*eNOS* and *VEGF*	RT-PCR	[107]
STZ-induced Swiss Albino mice	N/A	411 upregulated and 458 downregulated genes at E13.5	Affymetrix Mouse Genome 430 2.0 microarrays (GSE32078)	[102,108]
		802 upregulated and 1295 downregulated genes at E15.5		
STZ-induced C57BL/6 mice	N/A	*Isl1, Tbx1, Tbx20, Fgf10, Mef2c, Nkx2-5,* and *Hand2*	scRNA-Seq	[77]
STZ-induced C57BL/6 mice	VSD, ASD, AVSD, TGA, DORV and TOF	*Gata4, Gata5,* and *Vegfa*	RT-PCR	[109]
STZ-induced Sprague Dawley rats	OFT defects	*Vegf*	In situ hybridization	[110]
STZ-induced C57BL/6 mice	ASD, VSD, AVSD, DORV, pulmonary valve stenosis, aortic valve stenosis, HLHS, HRHS	*Gata4, Hif1a, Cyclin D1, Notch1* and *Snail1*	RT-PCR	[111]
Glucose treated fertilized chicken eggs		*Glut1, p21* and *Cyclin D1*	RT-PCR	[112]
STZ-induced C57BL/6 mice	PTA, VSD, right sided aortic arch	*Cyclin D1, Cyclin D3, p21, p27, BMP4, Nkx2.5,* and *GATA4*	RT-PCR	[113]
STZ-induced C57BL/6 mice	VSD, PTA	*sFRP1, Dkk1, β-catenin, Islet1, Gja1, Versican, Wnt5a, NFAT2/4, Mrtf-b, Tpm1,* and *Rcan1*	RT-PCR	[114]
STZ-induced C57BL/6 mice		*TGFβ1, TGFβ3, TβRII, Smad2/3, Snai2, CTGF,* and *GDF1*	RT-PCR	[115]
HFD-induced C57BL/6 mice	PTA, VSD	*BiP, CHOP, calnexin, PDIA, GRP94,* and *XBP1*	RT-PCR	[116]
STZ-induced C57BL/6 mice	Aortic and pulmonary stenosis, PTA	24 genes of the WNT pathway were differentially expressed	RT-PCR	[117]
STZ-induced C57BL/6 mice	HRHS, heterotaxia, thin myocardial wall, endocardial cushions defect, VSD	22 genes of the TGFβ signaling were differentially expressed	RT-PCR	[118]
STZ-induced FVB mice	N/A	1024 differentially expressed genes on E8.5, and 2148 genes on E9.5	SOLiD SAGE mRNA deep sequencing (PRJNA275285)	[119]
		*Ndufa6, Actg1, Glut4, Fgf2, Tpm4, Marcksl1, Myosin1H, Axin1, Mrto4, Psmc4, Ptpn11, Mrps23, Rnaseh2c, Cdk1* and *Ubxn8*	RT-PCR	
STZ-induced FVB mice	OFT defects	*Pax3*	LacZ reporter assay	[120]
STZ-induced Sprague Dawley rats	N/A	*Nppa, Nppb, Myh2, Myh3, Atp2a2, Kcnip2, Ucp2/3, Slc2a4, Egln3,* and *Tnfrsf12a*	RT-PCR	[88]
STZ-induced Sprague Dawley rats	N/A	*CuZnSOD, MnSOD, Gpx-1, AR, p53, PARP, Shh, Ret, Bmp-4, VEGF-A,* and *TNF-α*	RT-PCR	[121]

## Data Availability

All the data used to write this manuscript are presented in the text, tables, or references.

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
