# Peer review of "Maternal Pre-Existing Diabetes: A Non-Inherited Risk Factor for Congenital Cardiopathies"

_ijms, 2023, doi:10.3390/ijms242216258_

Round 1

Reviewer 1 Report

Comments and Suggestions for Authors

The manuscript by Ibrahim and colleagues focuses on maternal preexisting Diabetes and its—impact on congenital heart defects. The comprehensive review includes figures and tables supporting the text's information. The flow and organization are good, and the topic discussed should interest developmental biologists and clinical researchers with an emphasis on metabolic conditions. Therefore, I am supportive of accepting the manuscript; however, the authors should address the concerns of this reviewer before the manuscript can be accepted for publication.

1-   Table 2 presents a list of genes critical for cardiac development and anomalously expressed after exposure to the hyperglycemic milieu and the model from which data were generated. The table should be referenced in the main text.

2-   In the same line, are there correlations to GWAS or other large datasets of human CHD and Diabetes wherein differential gene expression or SNP for genes listed in Table 2 were also detected? The data could be added to Table 2 or the main text.

3-   Authors should discuss how the findings in experimental models (both in vivo and in vitro) can shed light on potential treatments to prevent or mitigate CHD in the offspring of diabetic mothers.

4-   While it is not the primary focus of the manuscript, is CHD presented in association with other congenital defects? Specifically, organs wherein the contribution of the second heart field is necessary for organ formation.

Author Response

We thank the reviewer#1 for the constructive assessment of our manuscript, the support, and the suggestions to improve it. Below, a point-by-point response to each of the comments is given.

1-   Table 2 presents a list of genes critical for cardiac development and anomalously expressed after exposure to the hyperglycemic milieu and the model from which data were generated. The table should be referenced in the main text.

We have referenced this table in the main text (line 383, section "Molecular mechanisms underlying the teratogenic effect of maternal pregestational diabetes").

2-   In the same line, are there correlations to GWAS or other large datasets of human CHD and Diabetes wherein differential gene expression or SNP for genes listed in Table 2 were also detected? The data could be added to Table 2 or the main text.

We agree with the reviewer, this is an important point that should be mentioned. We added in the section “Heightened CHD risk due to combined effect of genetic predisposition and hyperglycemia” the following sentences:

A retrospective study demonstrated that maternal diabetes, NKX2.5 variants, and their interactions were significantly associated with CHD in offspring (PMID: 33062711). The interaction between these two factors deserves further investigation, especially in relation to the specific subtypes of CHD involved (lines 553-556).

Genetic susceptibility can be influenced by environmental factors, and conversely, environmental factors can also be influenced by genetic factors (PMID: 28978192). A recent study underscores the necessity of exploring the mechanisms that link CHD risk with modifiable risk factors (PMID: 35130025). The authors conclude that genome-wide de novo variants are not associated with maternal diabetes or obesity in cases of congenital heart disease. This implies that other factors, such as inherited genetic variants, the metabolic influences of the mother on the developing heart, or environmental factors, should be considered as crucial areas for future research to gain a better understanding of their impact on CHD risk. It’s worth noting that numerous pathogenic CHD variants display varying penetrance and expressivity, suggesting that environmental factors could potentially modify the severity of CHD. Consistent with this, mouse models have demonstrated that the penetrance of NOTCH1-related CHD is heightened when exposed to gestational hypoxia (PMID: 31813956, lines 564-574).

3-   Authors should discuss how the findings in experimental models (both in vivo and in vitro) can shed light on potential treatments to prevent or mitigate CHD in the offspring of diabetic mothers.

Pre-existing diabetes unquestionably exerts an adverse impact on both pregnancy and the cardiac development of the fetus, even when women maintain adequate glycemic control. With the growing number of women affected by this condition in recent times, it is crucial to adopt a proactive approach when it comes to educating, preventing, and managing the metabolic health of these patients. There is a crucial need to pursue research efforts in this domain to gain comprehensive insights into the multifaceted connection between maternal diabetes and fetal cardiac abnormalities. Additionally, genetic factors may amplify susceptibility to specific environmental influences, a phenomenon that necessitates clarification. There are ongoing investigations into potential treatments for pregnant women with diabetes. Future studies using complementary experimental models should assess whether these potential treatments effectively reduce the risk of CHDs in offspring exposed to them. These experimental models will help pinpoint essential regulatory checkpoints throughout multiple stages of cardiac development, leading to uncover why infants exposed to teratogenic agents like hyperglycemia are more susceptible to fetal cardiac developmental issues. By utilizing dynamic methodologies, single-cell transcriptomics, epigenetic investigations, and lineage analysis we will improve our comprehension of how maternal hyperglycemia affects these stages and how potential treatments may reduce the risk of CHDs in offspring. This will facilitate early and efficient interventions as well as the development of effective therapies.

This commentary is included in the revised manuscript (lines 283-297).

4-   While it is not the primary focus of the manuscript, is CHD presented in association with other congenital defects? Specifically, organs wherein the contribution of the second heart field is necessary for organ formation.

We thank the reviewer for this suggestion. We added in the revised manuscript (lines 283-297) the following sentences:

The progenitors of the second heart field have been shown to contribute to both the heart and skeletal muscles of the head and neck. They give rise to not only cardiomyocytes but also endothelial, endocardial, and smooth/skeletal muscle cells (PMID: 34798131, PMID: 32014863, PMID: 20457151). Since hyperglycemia during pregnancy is associated with various congenital defects, exploring the association between a CHD type and congenital defects affecting the head and neck muscle lineages in patients could yield significant insights. Another significant aspect along these lines is that CHD is frequently accompanied by genetic syndromes presenting both cardiac and extra-cardiac anomalies (PMID: 27234354, PMID: 35645294, PMID: 34272501). The DiGeorge syndrome (22q11.2 deletion) is a striking example as it is associated with prevalent cardiac malformations and craniofacial defects. Conotruncal lesions such as interrupted aortic arch, truncus arteriosus, tetralogy of Fallot, and ventricular septal defects are frequently diagnosed in children with 22q11 deletion. The gene TBX1 located on chromosome 22q11.21 has been found in patients with the DiGeorge syndrome. Since Tbx1 is expressed in the second heart field progenitor cells and is altered by maternal diabetes (PMID: 35970860), the prevalence of TBX1-associated CHD may correlate with craniofacial defects upon exposure to maternal hyperglycemia. Up to date, there have been no documented reports on this association in either human clinical data or mouse studies.

Reviewer 2 Report

Comments and Suggestions for Authors

The Authors focused on a review of the maternal pre-existing diabetes: a non-inherited risk factor for congenital cardiopathies. This is an interesting topic and a systematic and comprehensive review. The manuscript is well structured, but there are some important facts that authors should add.  

The title clearly describes the articleThe Abstract presents an accurate description of the case and its implications. In the introduction, the Authors describe exactly what they wanted to achieve and clearly state the problem being investigated.

Key points to consider:

The Authors missing methodology section which is, in my opinion, very important for a clear structure of review.

The methodology section should provide information of literature searching process. If not, is not adequate for review form. Please add details about methodology of this study:

How was the data collected?

What databases the authors used to search for literature?

What were the search guidelines? Keywords? Time of publication?

How many articles were rejected and for what reason?

Please provide flow chart of searching.

Author Response

Our manuscript strives for a thorough examination of the existing literature and aims to serve as a state-of-the-art review. The reviewer's comments mainly concern a meta-analysis and are outside the scope of our manuscript.